# Estimated Oxygen Consumption with the Abbreviated Method and Its Association with Vaccination and PCR Tests for COVID-19 from Socio-Demographic, Anthropometric, Lifestyle, and Morbidity Outcomes in Chilean Adults

**DOI:** 10.3390/ijerph19116856

**Published:** 2022-06-03

**Authors:** Jaime Vásquez-Gómez, César Faúndez-Casanova, Ricardo Souza de Carvalho, Franklin Castillo-Retamal, Pedro Valenzuela Reyes, Yeny Concha-Cisternas, Pablo Luna-Villouta, Cristian Álvarez, Andrés Godoy-Cumillaf, Claudio Hernández-Mosqueira, Igor Cigarroa, Alex Garrido-Méndez, Carlos Matus-Castillo, Marcelo Castillo-Retamal, Ivana Leao Ribeiro

**Affiliations:** 1Centro de Investigación de Estudios Avanzados del Maule (CIEAM), Universidad Católica del Maule, Talca 3460000, Chile; jvasquez@ucm.cl; 2Laboratorio de Rendimiento Humano, Universidad Católica del Maule, Talca 3460000, Chile; cfaundez@ucm.cl (C.F.-C.); rsouza@ucm.cl (R.S.d.C.); fcastillo@ucm.cl (F.C.-R.); 3Departamento de Ciencias de la Actividad Física, Facultad de Ciencias de la Educación, Universidad Católica del Maule, Talca 3460000, Chile; 4Área de Actividad Física y Deportes, Técnico en Deportes, Centro de Formación Técnica Santo Tomás, Rancagua 2820000, Chile; pvalenzuelar@santotomas.cl; 5Escuela de Kinesiología, Facultad de Salud, Universidad Santo Tomás, Talca 3460000, Chile; yenyconchaci@santotomas.cl; 6Pedagogía en Educación Física, Facultad de Educación, Universidad Autónoma de Chile, Talca 3460000, Chile; 7Facultad de Educación, Pedagogía en Educación Física, Universidad San Sebastián, Concepcion 4030000, Chile; pablo.luna@uss.cl; 8Programa de Doctorado en Ciencias de la Actividad Física, Facultad de Ciencias de la Educación, Universidad Católica del Maule, Talca 3460000, Chile; 9Exercise and Rehabilitation Sciences Laboratory, School of Physical Therapy, Faculty of Rehabilitation Sciences, Universidad Andres Bello, Santiago 7591538, Chile; cristian.alvarez@unab.cl; 10Facultad de Educación, Pedagogía en Educación Física, Universidad Autónoma de Chile, Temuco 4780000, Chile; andres.godoy@uautonoma.cl; 11Departamento de Educación Física, Deportes y Recreación, Universidad de La Frontera, Temuco 4780000, Chile; claudiomarcelo.hernandez@ufrontera.cl; 12Escuela de Kinesiología, Facultad de Salud, Universidad Santo Tomás, Los Angeles 4440000, Chile; icigarroa@santotomas.cl; 13Departamento de Ciencias del Deporte y Acondicionamiento Físico, Universidad Católica de la Santísima Concepcion, Concepcion 4030000, Chile; agarrido@ucsc.cl (A.G.-M.); cmatus@ucsc.cl (C.M.-C.); 14Departamento de Kinesiología, Facultad de Ciencias de la Salud, Universidad Católica del Maule, Talca 3460000, Chile; 15Escuela de Ciencias del Deporte y Actividad Física, Facultad de Salud, Universidad Santo Tomás, Talca 3460000, Chile

**Keywords:** cardiorespiratory fitness, SARS-CoV-2, vaccines, polymerase chain reaction, adult

## Abstract

COVID-19 causes cardiovascular and lung problems that can be aggravated by confinement, but the practice of physical activity (PA) could lessen these effects. The objective of this study was to evaluate the association of maximum oxygen consumption ($\dot{V}$O_2_max) with vaccination and PCR tests in apparently healthy Chilean adults. An observational and cross-sectional study was performed, in which 557 people from south-central Chile participated, who answered an online questionnaire on the control of COVID-19, demographic data, lifestyles, and diagnosis of non-communicable diseases. $\dot{V}$O_2_max was estimated with an abbreviated method. With respect to the unvaccinated, those who received the first (OR:0.52 [CI:0.29;0.95], *p* = 0.019) and second vaccine (OR:0.33 [CI:0.18;0.59], *p* = 0.0001) were less likely to have an increased $\dot{V}$O_2_max. The first vaccine was inversely associated with $\dot{V}$O_2_max (mL/kg/min) (β:−1.68 [CI:−3.06; −0.3], *p* = 0.017), adjusted for BMI (β:−1.37 [CI:−2.71; −0.03], *p* = 0.044) and by demographic variables (β:−1.82 [CI:−3.18; −0.46], *p* = 0.009); similarly occur for the second vaccine (β: between −2.54 and −3.44, *p* < 0.001) on models with and without adjustment. Having taken a PCR test was not significantly associated with $\dot{V}$O_2_max (mL/kg/min). It is concluded that vaccination significantly decreased $\dot{V}$O_2_max, although it did not indicate cause and effect. There is little evidence of this interaction, although the results suggest an association, since $\dot{V}$
O_2_max could prevent and attenuate the contagion symptoms and effects.

## 1. Introduction

As in other countries of the world, Chile is carrying out the control process of SARS-CoV-2 (Severe Acute Respiratory Syndrome Coronavirus), also called COVID-19 (Coronavirus Disease), regarding the taking of PCR (Polymerase Chain Tests Reaction) and vaccination. On the one hand, the positivity of the PCR test has been less than 3% in the country, with about 1375 new cases daily and almost 51,000 tests administered in mid-December 2021 [1]. On the other hand, and the same date, the population that received the first two doses of the vaccine has exceeded 90% at the national level [2], and in the central-southern macro zone of Chile (regions of O’Higgins, Maule, Ñuble, and Biobío) [3], more than 90% of people over 18 years of age have had two doses of the vaccine [4]. These figures are being updated weekly by the pertinent institutions of the Republic of Chile mentioned recently.

It has been suggested that the coronavirus family carries respiratory diseases that include various symptoms [5] and that the SARS virus (Severe Acute Respiratory Syndrome), which is the predecessor of the current COVID-19, compromises the cardiovascular system, causing some alterations [6], and decreases the ability to perform physical exercise [7]. This COVID-19 virus causes cardiovascular problems [6,8] with respiratory symptoms after an acute period of contagion [8] and after several months of contracting the virus [9], with limitations on the performance of exercise physical in the post-hospitalization phase [10]. Even cardiovascular disorders due to COVID-19 have been linked to hospitalization and death [8]. Specifically, this virus activates signals in a cascade that produce inflammation at the lung level. It also causes mitochondrial dysfunction, which decreases its biogenesis, reduces immunity, ATP resynthesis, and increases oxygen reactive substances such as free radicals [11], although it has been postulated that the regular practice of PA acts inversely on these phenomena produced by COVID-19 [11], reducing the effects of viral infections, systemic inflammation [12,13], the risk of hospitalization, admission to the intensive care unit, and death from COVID-19 [13]. However, sanitary confinement measures have gone against PA [8,14], since there has been a decrease in the time allocated to its practice, causing a decrease in strength, muscle mass, and cardiorespiratory fitness (CRF) [15], including chronic stress that affects mental health [16]. Confinement could lead to increases in sedentary lifestyle [8], which can also affect body adiposity and CRF [14], which in long periods of rest has progressively reduced $\dot{V}$O_2_max in young adults [17]. It has even been postulated that a sedentary life increases the probability of death due to the virus [12]. The practice of PA and the development of CRF is relevant due to the repercussions on risk factors and the immune system [18]; therefore, it is necessary to apply cardiopulmonary stress tests for clinical and clinical research control [8].

One of the ways to evaluate CRF is through abbreviated methods that use demographic data, body adiposity, lifestyles, and cardiometabolic diseases, but predict CRF without performing physical exercise [19]. Several studies have used abbreviated methods over time that have been validated in different demographic contexts and with varied morpho-functional characteristics and life habits of the population [20,21,22,23,24,25,26,27]. As these become relevant and feasible for larger-scale population research due to their simplicity, low cost, time, supplies, trained personnel, etc., especially in during the current pandemic, where sanitary restrictions make it very difficult to collect empirical data, and evaluation through direct methods (gold standard) or field tests are not feasible [19]. The appearance of COVID-19 has brought with it elements to which societies have had to adapt quickly. Because of the pandemic, it has been necessary for public health bodies to generate evidence on the variables that could influence the contagion and the variables of PA. In turn, the results of this can be transferred to the Chilean health system and materialize from primary care to tertiary care. Governments can establish guidelines for PA during the pandemic period [28], ensure necessary PA counseling in diverse health centers [29] and that popular PA programs are implemented [30], to complement the methods of prevention and treatment of the virus and its current variants. The importance of active participation of the population in adherence to the PA practice should also be taken into account.

There has been a scarce evaluation of CRF as well as other aspects of physical fitness studies in population before the COVID-19 pandemic [18], the relevance of the evaluation of cardiopulmonary function [8], and a higher CRF has been associated with a lower risk of positivity in the PCR test [31]. Thus, data need to be collected and given practical use in an eventual “post pandemic” stage. The objective of this research was to evaluate the association of $\dot{V}$O_2_max with vaccination and PCR tests against COVID-19 according to socio-demographic variables in apparently healthy people over 18 years of age in the central-southern macro zone of Chile.

## 2. Materials and Methods

This was a cross-sectional observational study based on the recommendations of the Strengthening the Reporting of Observational Studies in Epidemiology (STROBE) Statement [32] with a non-random and convenience sample (non-probabilistic/purposive sampling) that was captured through virtual social networks (massive messages by WhatsApp (Meta Platforms, Inc., Menlo Park, CA, USA, support@whatsapp.com), Instagram (Meta Platforms, Inc., San Francisco, CA, USA, support@instagram.com). Participants who consented to give their personal information had to complete a one-time online self-report questionnaire, which was applied between May and August 2021. They were asked to give their consent, declare if they were 18 years old or older, and if they reside in one of the four regions of the central-southern macro zone of Chile: O’Higgins, Maule, Ñuble, and Biobío. If someone did not meet the inclusion criteria, the questionnaire was automatically closed without the participant being able to complete it; therefore, duplicated answers were not possible, avoiding selection bias. The final sample was made up of 557 participants of Chilean nationality (54% women) aged 28.9 ± 9.7 years (Figure 1).

The study was approved by a scientific ethics committee and all participants gave their virtual consent to access the questionnaire anonymously, in compliance with Law No. 19,628 of the Republic of Chile, regarding the protection of personal data. The online questionnaire was guided by the International Ethical Guidelines for Health-Related Research with Human Beings on the “Use of data obtained in online environments and digital tools in health-related research” and “Research in disaster situations and disease outbreaks”, both prepared by the Council for International Organizations of Medical Sciences [33] in collaboration with the World Health Organization.

The Google Forms platform was used for the questionnaire, in which anthropometric data (weight, height, and BMI [body mass index] [34]) were collected. Ad hoc questions were used to obtain information on inoculation against COVID-19 (first and second vaccines: SINOVAC, PFIZER, CANSINO, ASTRAZENECA), performances and results of any PCR test, demographic data (sex, age, area of residence, educational level, marital status, and occupational situation), information on lifestyle habit, walking pace, monthly PA and sitting time, and diagnosis of metabolic diseases (hypertension, diabetes, high cholesterol, heart attack, vascular accident, or cerebral thrombosis) [35]. The $\dot{V}$O_2_max was estimated in absolute terms (L/min) using an abbreviated method through the variables of body weight, age, and sex [26,36,37] and it was transformed to its relative form (mL/kg/min) by multiplying the absolute value by 1000 mL; this product was divided into body weight [38]. The relative form of $\dot{V}$O_2_max was classified according to sex and age [39] into two categories: “very low, low, or normal” or “good, excellent, or superior”.

The data were presented in mean values and standard deviation, which were subjected to a Kolmogorov–Smirnov normality test, and the variables between both sexes were compared with a Student’s *t*-test or Kruskal–Wallis test, as appropriate. In addition, the categorical variables were presented as absolute and relative values, and the prevalence was determined according to the sex of the participants with the chi-square test (*x*^2^) or Fisher’s exact test. The continuous and categorical variables were accompanied by their respective confidence intervals (95% CI). Finally, it was evaluated to what extent COVID-19 control influenced $\dot{V}$O_2_max; for this, the increase or decrease in oxygen consumption associated with vaccination and the PCR test was evaluated through the probability odds ratio (OR) calculation, and using linear regression with the beta (β) coefficient accompanied by unadjusted regression models adjusted for BMI, demographic variables, lifestyle variables, and the diagnosis of cardiometabolic diseases. All the analysis was carried out with the STATA v.14 program considering statistical significance with a *p*-value < 0.05 (38 cases that had erroneous and/or incomplete records were not part of the analysis, in addition, 10 cases that decided not to participate in the study, leaving a total of 557 participants).

## 3. Results

Table 1 shows the characteristics of the sample. It was observed that there were differences between men and women in $\dot{V}$O_2_max, either in absolute or relative terms, and that this variable was categorized as “low, very low, or normal” in a high percentage (~80%), both in men as in women, although there was no prevalence by either of the two sexes. It was also noted that a high percentage of respondents reported having been vaccinated against COVID-19 with the first (~80%, between January and July 2021) and second dose (~70%, between February and August 2021). The SINOVAC brand (CoronaVac) was the one with the highest application (58% of the total) and significant differences in prevalence were found between men and women. Regarding the PCR test, 57.3% of the total participants underwent this preventive and diagnostic test, with there being differences by sex, and the “negative” result was the one with the highest percentage (~90%). Finally, lifestyle habits such as sitting time, walking pace, and PA practice showed differences between men and women, while the diagnosis of metabolic and cardiovascular diseases had a low percentage (~1 to 20%). The prevalence of diabetes was significantly marked by sex, which could be attributed to the fact that most of the participants were young adults.

People who were vaccinated against COVID-19, either with the first or second dose, were significantly less likely (48 and 67%, respectively) to have a higher $\dot{V}$O_2_max compared to people who were not vaccinated. For their part, those who underwent the PCR test were less likely to have better oxygen consumption compared to those who did not undergo the test. Those who obtained a positive result on the PCR test were more likely to have a higher $\dot{V}$O_2_max compared to those that obtained a negative result, although the latter results were not statistically significant (Table 2).

Table 3 and Table 4 show the control variables for COVID-19, of which some of them were able to predict $\dot{V}$O_2_max. In Table 3, no variable was significantly associated with absolute oxygen consumption (L/min), except for the model adjusted for BMI (β: −0.24 [CI: −0.37; −0.11], *p* < 0.001), in which all had a negative interaction. The effect of being vaccinated for the first or second time, performing a PCR test, and having a negative result in the latter decreased the $\dot{V}$O_2_max. For its part, for oxygen consumption in relative terms, the first vaccine decreased it by −1.68 mL/kg/min in the unadjusted model: by −1.37 mL/kg/min when adjusted for BMI and by −1.82 mL/kg/min when adjusted for demographic variables. The second dose of the vaccine had significant associations with a decrease in $\dot{V}$O_2_max (β between −2.54 and −3.44 mL/kg/min) in the unadjusted model and all the adjusted models (Table 4).

## 4. Discussion

The most important result of this research was that vaccination against COVID-19 had an inverse association with $\dot{V}$O_2_max. People who had received first and second doses had a decrease in their CRF compared to those who did not vaccinate. In this association variables of body adiposity, demographics, lifestyles, and diagnosis of cardiometabolic diseases intervened. It has been suggested that a small change in $\dot{V}$O_2_max could limit the risk of contagion [40] and that it would be expected that the CRF would decrease during the acute stage of COVID-19 infection [8]. Accordingly, we suggest as a hypothesis that the vaccine produced an acute effect of lowering $\dot{V}$O_2_max since the people who reported having been vaccinated were inoculated with the COVID-19 virus itself. We can support this hypothesis based on the findings of Batatinha et al. [41], who, when applying a pre-test, vaccine (Pfizer and Johnson&Johnson), and post-test, did not find significant differences in $\dot{V}$O_2_max on a cycle ergometer between the two measurements, both in infected, non-infected (both vaccinated), and control participants (not vaccinated), although the sample was only 12 participants.

We can also hypothesize that the same subjects who reported have been vaccinated in the present study perceived a diminished state of health. For that reason, they went to health centers to be vaccinated, due to being motivated by confinement [8,14], fear and possibly misinformation about the pandemic [16], decreased practice of PA and physical fitness [15], changes in body fat [14], and increases in sedentary lifestyle [8]. It should be considered that so far, there have been no reports regarding the impact of the vaccine (SINOVAC, for example, which was the most reported in this research) on $\dot{V}$O_2_max, since there is a knowledge gap to corroborate what reports our results.

The comparative evidence indicates that lung capacity and volume were significantly protective variables of severe symptoms due to COVID-19 (relative risk (RR) < 1.0; *p* < 0.05). Moreover, “fast” walking speed was a significant protective variable of excess body adiposity, reflected by the BMI, while conversely, having a “slow” walking pace was a risk factor [42]. Another study reinforced this result on gait speed, reporting that subjects who walked at a moderate intensity reduced the probability of hospitalization due to COVID-19 by 64% (OR: 0.36 [CI: 0.13; 0.98], *p* = 0.04) compared to those who walked slowly [43], so a slow walking pace can be a predictor of serious contagion by COVID-19 [44]. These findings could explain our results, since among participants who declared that they had received the first vaccine, and when this was adjusted for BMI, a negative and significant association was demonstrated with the relative $\dot{V}$O_2_max (Table 4).Being vaccinated a second time was inversely associated with oxygen consumption, both absolute (Table 3) and relative (Table 4). The model that considered the second vaccination and that was adjusted for lifestyle variables, which included walking pace or speed, also showed an inverse and significant association with relative oxygen consumption (Table 4). This is in opposition to previous literature. Previous research has also reported that age, bodyweight loss, being an active smoker, and length of hospitalization for COVID-19 were negatively and significantly associated with the prediction of $\dot{V}$O_2_max [7]. Although our results demonstrated that only the second dose of the vaccine was negatively associated with $\dot{V}$O_2_max when it was adjusted for lifestyle habits, among them the smoking habit (Table 4).

The evidence regarding the PCR test is still limited. One study has shown that people with a “moderate” (RR: 0.93 [CI: 0.72; 1.21]) or “high” CRF (RR: 0.77 [CI: 0.52; 1.15]) had a lower relative risk of positivity in the said test compared to those who had a low CRF level, although not significantly [31]. On the contrary, our results indicated that the positivity of the PCR test was associated with a greater probability of increasing $\dot{V}$O_2_max, although this was not statistically significant either (Table 2).

It has been suggested that non-serious, hospitalized patients with COVID-19 have had a lower $\dot{V}$O_2_max at 3 months after discharge or convalescence [8]. At the same time, it has also been found that the probability of hospitalization for COVID-19 was higher in patients who had lower CRF, and, conversely, it was lower in people with a higher level of fitness, so lower CRF was 2.88 times more likely to be hospitalized (OR: 3.88 [CI: 1.78; 8.77]) [45]. These same researchers also demonstrated that the probability of hospitalization was lower in those younger than 65 years compared to those older this age, in non-obese than in obese, and in men than in women, although there were no significant differences between any of them. Some chronic pathologies, such as diabetes, kidney disease, coronary arteries, heart failure, cancer, and hypertension, had a significant association (OR: from 1.95 to 5.39; *p* < 0.05) with the probability of hospitalization due to COVID-19, but when these variables were adjusted by CRF, five of these pathologies were no longer associated [46]. It has even been shown that after 4 months of follow-up, for patients hospitalized for COVID-19, those who were on mechanical ventilation showed significantly less cardiopulmonary capacity (*p* < 0.05) and shorter predicted distance (*p* < 0.05) in the 6-min walk test compared to patients that did not require ventilation [47]. The distance in the walk test was also significantly lower in patients who were hospitalized compared to control subjects [48]. A prospective study has shown that subjects with “medium” and especially “high” CRF had significantly lower probabilities of hospitalization due to COVID-19 (OR: 0.76 [CI: 0.67; 0.85]), admission to the intensive care unit (OR: 0.61 [CI: 0.48; 0.78]), and mortality due to the virus (OR: 0.56 [CI: 0.37; 0.85]) compared to subjects with a low CRF level [18]. In this last and delicate aspect, a low CRF also resulted in a significantly 134% higher risk of mortality from COVID-19 (RR: 2.34 [CI: 1.35; 4.05]) when compared to moderate- or high-level fitness [31]. It has also been found that patients who were hospitalized for COVID-19 had significantly lower $\dot{V}$O_2_max in a cycle ergometer test, compared to control subjects [48] and that mechanically ventilated patients had limited exercise capacity due to the decrease in lung capacity and peripheral muscle mass [49]. Our research did not include questions about the hospitalization that the respondents may have had, which could be considered sensitive information and perceived as invasive; moreover, access to these records is restricted and they belong to patients and health centers managed by the Republic of Chile. However, it is pertinent to demonstrate and discuss the importance of $\dot{V}$O_2_max concerning hospitalization, due to the implications that it may have in terms of prevention and possible treatment of the sequelae of COVID-19.

Finally, the critical literature expresses that $\dot{V}$O_2_max should be considered a variable of vital signs [8,40], at the beginning of the clinical evaluation, together with demographic and other data, since COVID-19 has challenged us to think about complementary forms of evaluation. Considering that a small increase in $\dot{V}$O_2_max is likely to have benefits in the body, and can serve to discriminate between patients with higher and lower risk of contagion [40]. For its part, regarding the evaluation of $\dot{V}$O_2_max through stress tests, it is necessary to apply them during the virus contagion stage to obtain more information about these tests and how they could help control COVID-19. Since to date, there are no specific records of stress tests for this virus [10] from they have been carried out on very small samples [41]. On the other hand, it is known that the development of aerobic exercise of moderate or intense intensity increases lymphocytes and other immune cells in the blood [5]; therefore, one of the utilities that the evaluation of CRF status could have in the young adult age is that it could influence the severity of COVID-19 many years later [18]. There is even evidence that has shown significant decreases in $\dot{V}$O_2_max before and after an outbreak of COVID-19 in young convalescents, and at the same time, a statistically lower $\dot{V}$O_2_max compared to newly infected and asymptomatic individuals [50]. The development of the CRF could materialize with popular PA programs that aim to improve public health [30].

A strength of this research is that this was a pioneering study in Chile on $\dot{V}$O_2_max and the control of COVID-19, in which $\dot{V}$O_2_max was evaluated through an abbreviated method, this being a novel and viable methodology to use in times of sanitary restriction and confinement measures. It highlights the contribution of the abbreviated methods for predicting $\dot{V}$O_2_max, encouraging this variable to be included in the clinical evaluation due to the implication it may have on COVID-19, and the prevalence in public health on chronic communicable and non-chronic diseases. Another strength is that this study was carried out in a context where the vaccination process has had coverage by a high percentage of the population. One limitation is that a self-report questionnaire was used, which could lead to participants underestimating or overestimating their responses. This condition was also described as a limitation in population studies where questionnaires and self-reports have been used [51,52]. However, questionnaires are instruments that have been used frequently to collect information during the pandemic in many places in the world [53,54], including Chile [52,55], to apply as a digital tool in disaster and disease outbreak conditions [33]. Finally, it should be considered that this was an observational cross-sectional study; thus, the results of the association between the $\dot{V}$O_2_max/control variables of COVID-19 did not indicate cause and effect, which is supported by other research that has used a cross-sectional design [52,56,57]. Future research could attempt to associate $\dot{V}$O_2_max with the hospitalization stage, in the first instance through self-reported data, by improving the questionnaire applied in this study, and later by accessing data released by health institutions, both private and public. Moreover, future papers should stimulate the development of the CRF by calculating the MET (metabolic equivalent of task), since these variables are linked to each other, and the CRF is associated with health outcomes (i.e., adiposity, lifestyles, and cardiometabolic diseases) [25].

## 5. Conclusions

It is concluded that vaccination against COVID-19 significantly decreased the chances of increasing the $\dot{V}$O_2_max of the participants. A decrease in absolute $\dot{V}$O_2_max (L/min) was influenced by variables such as BMI. A decrease in $\dot{V}$O_2_max in relative terms (mL/kg/min) was influenced by BMI, demographic variables, life habits, and diagnosis of non-transmissible diseases. The second vaccine had significant associations in the unadjusted model, as well as all the adjusted models. The PCR examination was not significantly associated with $\dot{V}$O_2_max. None of these associations demonstrated cause and effect.

## Figures and Tables

**Figure 1 ijerph-19-06856-f001:**
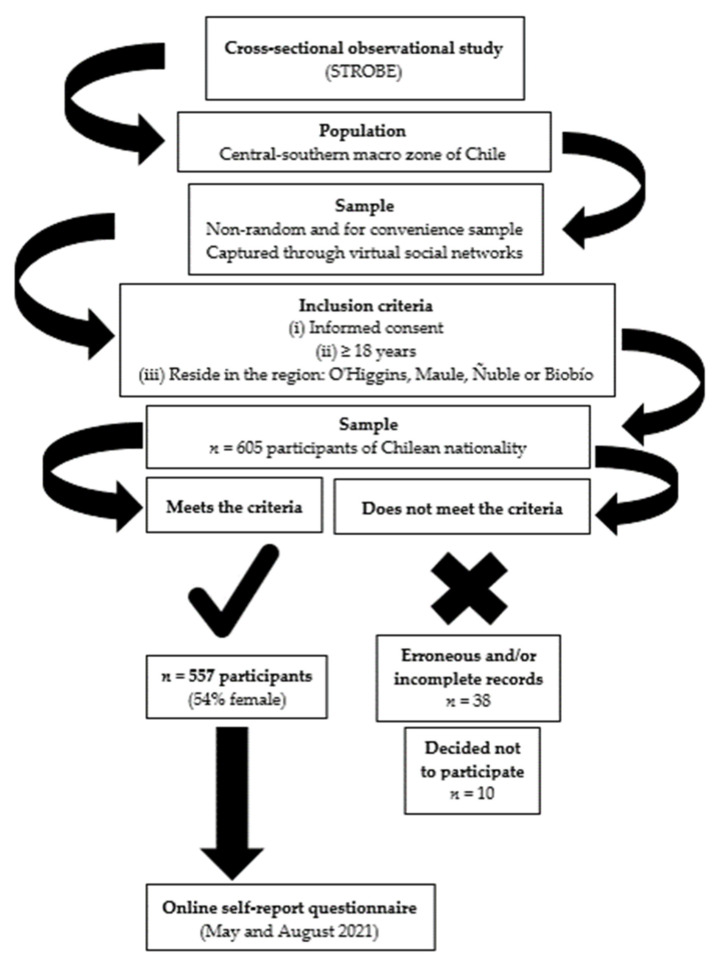
Flow diagram of actions.

**Table 1 ijerph-19-06856-t001:** Sample characteristics.

Variables	Total (557)	Male (256)	Female (301)	*p*-Value (a)
Mean	SD	CI	Mean	SD	CI	Mean	SD	CI
**Age (years)**	28.9	9.7	28.1; 29.8	28	9.3	26.8; 29.1	29.8	10.1	28.6; 30.9	**0.0166 t**
**Weight (kg)**	71.7	13.1	70.6; 72.8	77.6	13.1	76; 79.2	66.6	10.8	65.4; 67.8	**0.0001 k**
**Height (cm)**	166.7	8.7	166; 167.5	173.5	6.4	172.7; 174.3	161	6	160.3; 161.7	**0.0001 k**
**BMI (kg/m^2^)**	25.7	3.7	25.3; 26	25.7	3.6	25.2; 26.1	25.6	3.8	25.2; 26.1	0.5636 t
**BMI,** *n* **(%)**										0.333 *x*^2^
Normal	280 (50.3)	--	46.1; 54.4	123 (48.1)	--	41.9; 54.1	157 (52.2)	--	46.4; 57.7	
Overweight/Obese	277 (49.7)	--	45.5; 53.8	133 (51.9)	--	45.8; 58	144 (47.8)	--	42.2; 53.5	
$\dot{V}$ **O_2_max (L/min)**	2.48	0.71	2.42; 2.54	3.12	0.52	3; 3.18	1.93	0.24	1.91; 1.96	**0.0001 k**
$\dot{V}$ **O_2_max (mL/kg/min)**	34.4	6.4	33.8; 34.9	40.2	3.6	39.8; 40.7	29.4	3.6	29; 29.8	**0.0001 k**
$\dot{V}$ **O_2_max, *n* (%)**										0.08 *x*^2^
Low, very low, or normal	479 (86)	--	82.8; 88.6	213 (83.2)	--	78; 87.3	266 (88.4)	--	84.2; 91.5	
Good, excellent or higher	78 (14)	--	11.3; 17.1	43 (16.8)	--	12.6; 21.9	35 (11.6)	--	8.4; 15.7	
**Sitting (hours)**	6	3.3	5.7; 6.3	5.6	3.4	5.2; 5.9	6.3	3.4	6; 6.7	**0.0021 k**
**Area,** *n* **(%)**										0.62 *x*^2^
Urban	492 (88.3)	--	85.3; 90.7	228 (89.1)	--	84.5; 92.3	264 (87.7)	--	83.4; 90.9	
Rural	65 (11.7)	--	9.2; 14.6	28 (10.9)	--	7.6; 15.4	37 (12.3)	--	9; 16.5	
**Vaccine 1,** *n* **(%)**										0.965 *x*^2^
Yes	453 (81.3)	--	77.8; 84.3	208 (81.3)	--	75.9; 85.5	245 (81.4)	--	76.5; 85.4	
No	104 (18.7)	--	15.6; 22.1	48 (18.7)	--	14.4; 24	56 (18.6)	--	14.5; 23.4	
**Vaccine brand, *n* (%)**										**<0.001 f**
SINOVAC	281 (63)	--	58.4; 67.3	114 (55.6)	--	48.7; 62.2	167 (69.3)	--	63.1; 74.8	
PFIZER	128 (28.7)	--	24.6; 33	67 (32.7)	--	26.5; 39.4	61 (25.3)	--	20.1; 31.2	
CANSINO	23 (5.2)	--	3.4; 7.6	10 (4.9)	--	2.6; 8.8	13 (5.4)	--	3.1. 9	
ASTRAZENECA	14 (3.1)	--	1.8; 5.2	14 (6.8)	--	4; 11.2	0	--	0.0; 0.0	
**Vaccine 2,** *n* **(%)**										0.078 *x*^2^
Yes	319 (69.2)	--	64.8; 73.2	138 (65.1)	--	58.4; 71.2	181 (72.7)	--	66.7; 77.8	
No	142 (30.8)	--	26.7; 35.1	74 (34.9)	--	28.7; 41.5	68 (27.3)	--	22.1; 33.2	
**PCR,** *n* **(%)**										**0.03** *x* ^2^
Yes	319 (57.3)	--	53.1; 61.3	134 (52.3)	--	46.2; 58.4	185 (61.5)	--	55.8; 66.8	
No	238 (42.7)	--	38.6; 46.8	122 (47.7)	--	41.5; 53.8	116 (38.5)	--	33.1; 44.1	
**PCR results,** *n* **(%)**										0.816 *x*^2^
Negative	290 (89)	--	85; 91.9	123 (88.5)	--	81.9; 92.8	167 (89.3)	--	83.9; 93	
Positive	36 (11)	--	8; 14.9	16 (11.5)	--	7.1; 218	20 (10.7)	--	6.9; 16	
**Smoker,** *n* **(%)**										0.286 *x*^2^
Current smoker	119 (21.4)	--	18.1; 24.9	53 (20.7)	--	16.1; 26.1	66 (21.9)	--	17.5; 26.9	
Former smoker	110 (19.7)	--	16.6; 23.2	44 (17.2)	--	13; 22.3	66 (21.9)	--	17.5; 26.9	
Never smoked	328 (58.9)	--	54.7; 62.9	159 (62.1)	--	55.9; 67.8	169 (56.2)	--	50.4; 61.6	
**Walking pace,** *n* **(%)**										**0.039 f**
Slow	23 (4.1)	--	2.7; 6.1	5 (1.9)	--	0.8; 4.6	18 (5.9)	--	3.7; 9.3	
Normal	352 (63.2)	--	59; 67.1	161 (62.9)	--	56.7; 68.6	191 (63.5)	--	57.8; 68.7	
Hurried	182 (32.7)	--	28.8; 36.6	90 (35.2)	--	29.5; 41.2	92 (30.6)	--	25.6; 36	
**PA practice,** *n* **(%)**										**<0.001** *x* ^2^
Does not practice	144 (25.9)	--	22.3; 29.6	42 (16.4)	--	12.3; 21.4	102 (33.9)	--	28.7; 39.4	
Yes, < 4 times/month	63 (11.3)	--	8.9; 14.2	27 (10.6)	--	7.3; 14.9	36 (12)	--	8.7; 16.1	
Yes, 1–2 times/week	148 (26.6)	--	23; 30.4	61 (23.8)	--	18.9; 29.4	87 (28.9)	--	24; 34.3	
Yes, ≥ 3 times/week	202 (36.2)	--	32.3; 40.3	126 (49.2)	--	43.1; 55.3	76 (25.2)	--	20.6; 30.4	
**High pressure,** *n* **(%)**										0.189 *x*^2^
No, they never told me	446 (80.1)	--	76.5; 83.1	202 (78.9)	--	73.4; 83.4	244 (81.1)	--	76.2; 85.1	
Yes, one time	53 (9.5)	--	7.3; 12.2	30 (11.7)	--	8.3; 16.2	23 (7.6)	--	5.1; 11.2	
Yes, more than once	35 (6.3)	--	4.5; 8.6	12 (4.7)	--	2.6; 8	23 (7.6)	--	5.1; 11.2	
I don’t remember, I’m not sure	23 (4.1)	--	2.7; 6.1	12 (4.7)	--	2.6; 8	11 (3.7)	--	2; 6.4	
**Diabetes,** *n* **(%)**										**0.019 f**
No	515 (94.5)	--	92.2; 96.1	245 (96.8)	--	93.7; 98.4	270 (92.5)	--	88.8; 94.9	
Yes	30 (5.5)	--	3.8; 7.7	8 (3.2)	--	1.5; 6.2	22 (7.5)	--	5; 11.1	
**High cholesterol, *n* (%)**										0.067 f
No, they never told me	434 (77.9)	--	74.2; 81.1	207 (80.9)	--	75.5; 85.2	227 (75.4)	--	70.2; 79.9	
Yes, one time	69 (12.4)	--	9.8; 15.4	33 (12.9)	--	9.2; 17.6	36 (12)	--	8.7; 16.1	
Yes, more than once	35 (6.3)	--	4.5; 8.6	9 (3.5)	--	1.8; 6.6	26 (8.6)	--	5.9; 12.4	
I don’t remember, I’m not sure	19 (3.4)	--	2.1; 5.2	7 (2.7)	--	1.3; 5.6	12 (4)	--	2.2; 6.9	
**Heart attack, *n*** **(%)**										0.711 f
No	541 (99.6)	--	98.5; 99.9	250 (99.6)	--	97.2; 99.9	291 (99.7)	--	97.5; 99.9	
Yes	2 (0.4)	--	0.09; 1.4	1 (0.4)	--	0.05; 2.7	1 (0.3)	--	0.04; 2.4	
**Vascular accident or cerebral thrombus,** *n* **(%)**										0.448 f
No	543 (99.5)	--	98.3; 99.8	252 (99.2)	--	96.8; 99.8	291 (99.7)	--	97.5; 99.9	
Yes	3 (0.5)	--	0.1; 1.6	2 (0.8)	--	0.1; 3.1	1 (0.3)	--	0.05; 2.4	

(a): the difference between male and female; BMI: body mass index; CI: confidence interval; f: Fisher’s exact test; k: Kruskal–Wallis test; PA: physical activity; PCR: Polymerase Chain Reaction; SD: standard deviation; t: Student’s *t*-test; u: Mann–Whitney U test; V˙O_2_max: maximum oxygen consumption; *x*^2^; Chi-square.

**Table 2 ijerph-19-06856-t002:** Probability of changes in $\dot{V}$O_2_max * due to control of COVID-19.

Variable	OR	CI (95%)	*p*-Value
Vaccine 1 (yes)	0.52	0.29; 0.95	**0.019**
Vaccine 2 (yes)	0.33	0.18; 0.59	**0.0001**
PCR (yes)	0.66	0.40; 1.11	0.099
PCR (−)	0.62	0.22; 1.96	0.32
PCR (+)	1.61	0.51; 4.34	0.32

CI: confidence interval; OR: odds ratio; PCR: polymerase chain test reaction; (−): negative; (+): positive. * mL/kg/min.

**Table 3 ijerph-19-06856-t003:** COVID-19 control variables that predict absolute $\dot{V}$O_2_max.

Variable	β	*p*-Value	CI 95%
$\dot{V}$ **O_2_max (L/min)**		**Vaccine first**	
Model 1	−0.04	0.587	−0.19; 0.11
Model 2	−0.08	0.251	−0.22; 0.05
Model 3	−0.03	0.643	−0.18; 0.11
Model 4	−0.01	0.85	−0.16; 0.13
Model 5	−0.03	0.669	−0.19; 0.12
Model 6	−0.005	0.948	−0.16; 0.15
$\dot{V}$ **O_2_max (L/min)**		**Vaccine second**	
Model 1	−0.13	0.059	−0.27; 0.005
Model 2	−0.24	**<0.001**	−0.37; −0.11
Model 3	−0.1	0.173	−0.24; 0.04
Model 4	−0.12	0.072	−0.27; 0.01
Model 5	−0.11	0.144	−0.26; 0.03
Model 6	−0.08	0.288	−0.23; 0.07
$\dot{V}$ **O_2_max (L/min)**		**PCR**	
Model 1	−0.04	0.508	−0.16; 0.07
Model 2	−0.06	0.276	−0.17; 0.05
Model 3	−0.03	0.559	−0.15; 0.08
Model 4	−0.03	0.564	−0.15; 0.08
Model 5	−0.05	0.428	−0.17; 0.07
Model 6	−0.04	0.522	−0.16; 0.08
$\dot{V}$ **O_2_max (L/min)**		**PCR (−)**	
Model 1	−0.19	0.132	−0.44; 0.05
Model 2	−0.13	0.266	−0.36; 0.11
Model 3	−0.17	0.169	−0.43; 0.07
Model 4	−0.2	0.119	−0.45; 0.05
Model 5	−0.16	0.204	−0.42; 0.09
Model 6	−0.16	0.205	−0.42; 0.09

CI: confidence interval; PCR: polymerase chain reaction; V˙O_2_max: maximum oxygen consumption; mL/kg/min: milliliters/kilogram/minute; (–): negative. Note: Model 1 is not adjusted. Model 2 is adjusted by BMI. Model 3 is adjusted by region, area of residence, educational level, marital status, and occupational situation. Model 4 adjusted for smoking habit, walking pace, monthly physical activity, and sitting time. Model 5 is adjusted for hypertension, diabetes, high cholesterol, heart attack, vascular accident, and cerebral thrombosis. Model 6 was adjusted by models 2, 3, 4, and 5.

**Table 4 ijerph-19-06856-t004:** COVID-19 control variables that predict the relative $\dot{V}$O_2_max.

Variable	β	*p*-Value	CI 95%
$\dot{V}$ **O_2_max (mL/kg/min)**		**Vaccine first**	
Model 1	−1.68	**0.017**	−3.06; −0.30
Model 2	−1.37	**0.044**	−2.71; −0.03
Model 3	−1.82	**0.009**	−3.18; −0.46
Model 4	−1.09	0.106	−2.42; 0.23
Model 5	−1.3	0.062	−2.68; 0.06
Model 6	−1.15	0.094	−2.5; 0.19
$\dot{V}$ **O_2_max (mL/kg/min)**		**Vaccine second**	
Model 1	−3.44	**<0.001**	−4.68; −2.20
Model 2	−2.92	**<0.001**	−4.15; −1.69
Model 3	−3.05	**<0.001**	−4.30; −1.79
Model 4	−3.28	**<0.001**	−4.48; −2.08
Model 5	−2.92	**<0.001**	−4.19; −1.66
Model 6	−2.54	**<0.001**	−3.8; −1.28
$\dot{V}$ **O_2_max (mL/kg/min)**		**PCR**	
Model 1	−0.89	0.106	−1.99; 0.19
Model 2	−0.73	0.173	−1.78; 0.32
Model 3	−0.8	0.142	−1.87; 0.26
Model 4	−0.66	0.210	−1.71; 0.37
Model 5	−0.94	0.090	−2.03; 0.14
Model 6	−0.63	0.237	−1.68; 0.41
$\dot{V}$ **O_2_max (mL/kg/min)**		**PCR (-)**	
Model 1	−1.24	0.263	−3.43; 0.93
Model 2	−1.49	0.173	−3.65; 0.66
Model 3	−0.91	0.413	−3.11; 1.28
Model 4	−1.46	0.174	−3.56; 0.64
Model 5	−0.92	0.398	−3.07; 1.2
Model 6	−0.93	0.387	−3.05; 1.18

CI: confidence interval; PCR: polymerase chain reaction; V˙O_2_max: maximum oxygen consumption; mL/kg/min: milliliters/kilogram/minute; (–): negative. Note: Model 1 is not adjusted. Model 2 is adjusted by BMI. Model 3 is adjusted by region, area of residence, educational level, marital status, and occupational situation. Model 4 adjusted for smoking habit, walking pace, monthly physical activity, and sitting time. Model 5 is adjusted for hypertension, diabetes, high cholesterol, heart attack, vascular accident, and cerebral thrombosis. Model 6 was adjusted by models 2, 3, 4, and 5.

## Data Availability

Not applicable.

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
