# Peer review of "Estimated Oxygen Consumption with the Abbreviated Method and Its Association with Vaccination and PCR Tests for COVID-19 from Socio-Demographic, Anthropometric, Lifestyle, and Morbidity Outcomes in Chilean Adults"

_ijerph, 2022, doi:10.3390/ijerph19116856_

Round 1
Reviewer 1 Report
After the authors’ point-by-point response and the revised version, the paper is organized well and easy to follow. The overall quality of the study enhances the readability of the survey. While English is used generally correctly from a grammar and syntax point of view.
Author Response
Dear Editor, Dr. Federica Valeriani
Special Issue "Sport, Physical Activity and Health at Time of COVID-19"
Response letter to reviewers
In addition to greeting you, we thank the reviewers for the corrections made that have improved the manuscript ijerph-1654180 “Estimated oxygen consumption with abbreviated method and its association with vaccination and PCR tests for COVID-19 from socio-demographic, anthropometric, lifestyle, morbidity outcomes in Chilean adults”. We proceed to respond point by point to said corrections, which are recorded in this response letter and also with track changes in the manuscript.
Cordially
Ivana Leao Ribeiro.
Escuela de Ciencias del Deporte y Actividad Física, Facultad de Salud, Universidad Santo Tomas, Talca PC: 3460000, Chile. ileao@ucm.cl. Phone: +569-712413622.
Marcelo Castillo-Retamal.
Facultad de Ciencias de la Educación, Departamento de Ciencias de la Actividad Física, Universidad Católica del Maule, Talca, PC:3460000, Chile. mcastillo@ucm.cl. Phone: +56 71 298 6414.
Reviewer 1
After the authors’ point-by-point response and the revised version, the paper is organized well and easy to follow. The overall quality of the study enhances the readability of the survey. While English is used generally correctly from a grammar and syntax point of view.
Dear Reviewer. Thank you for his comments on the new (improved) version of the manuscript, in content and formal aspects.
Reviewer 2 Report
INTRODUCTION
Very comprehensive introduction. It addresses the subject by justifying the study to be carried out with recent research on the field under study.
MATERIALS AND METHODS
The sample used for this study was selected through social networks, without really knowing if, for example, the data provided by the subjects were correct, so the estimate of Vo2max is not adequate, as well as the weight and height data of each of the subjects. I think that, in order to be able to conclude with such important statements as they do in the summary and in the conclusions section, they must be sure of the answers obtained in the research. The researchers should clarify how they ensured that the data provided by the subjects were accurate.
RESULTS
Please clarify whether 99.6% of the sample would have suffered a heart attack, before or after the vaccine or having COVID-19, as well as 99.5% of the sample would have suffered a vascular accident. In this case, if they would have suffered such a problem before any of these events, I think it would be important to highlight this as a criterion for the selection of the sample, as we are talking about an almost full percentage.
Author Response
Dear Editor, Dr. Federica Valeriani
Special Issue "Sport, Physical Activity and Health at Time of COVID-19"
Response letter to reviewers
In addition to greeting you, we thank the reviewers for the corrections made that have improved the manuscript ijerph-1654180 “Estimated oxygen consumption with abbreviated method and its association with vaccination and PCR tests for COVID-19 from socio-demographic, anthropometric, lifestyle, morbidity outcomes in Chilean adults”. We proceed to respond point by point to said corrections, which are recorded in this response letter and also with track changes in the manuscript.
Cordially
Ivana Leao Ribeiro.
Escuela de Ciencias del Deporte y Actividad Física, Facultad de Salud, Universidad Santo Tomas, Talca PC: 3460000, Chile. ileao@ucm.cl. Phone: +569-712413622.
Marcelo Castillo-Retamal.
Facultad de Ciencias de la Educación, Departamento de Ciencias de la Actividad Física, Universidad Católica del Maule, Talca, PC:3460000, Chile. mcastillo@ucm.cl. Phone: +56 71 298 6414.
Reviewer 2
INTRODUCCTION. Very comprehensive introduction. It addresses the subject by justifying the study to be carried out with recent research on the field under study.
Dear Reviewer. Thank you for your review of the introduction section.
The sample used for this study was selected through social networks, without really knowing if, for example, the data provided by the subjects were correct, so the estimate of Vo2max is not adequate, as well as the weight and height data of each of the subjects. I think that, in order to be able to conclude with such important statements as they do in the summary and in the conclusions section, they must be sure of the answers obtained in the research. The researchers should clarify how they ensured that the data provided by the subjects were accurate.
Dear Reviewer. Thank you for his methodological considerations. The possible methodological bias was mentioned in the introduction, methodology and discussion section, in the latter when we refer to the limitations of the study.
On the one hand, we assume the limitation on the application of self-report questionnaires and the accuracy of the data provided, but at the same time we justify their use in times of disease outbreaks (line 451 – 459). In addition, before analyzing the information, we filtered the incomplete, erroneous and extreme data, so we were certain of the reliability of the data (line 180 – 183). On the other hand, we have justified the estimation of VO2max with the abbreviated method based on its viability in larger scale epidemiological studies (line 100 – 108), we have also argued about the reliability and validity of the abbreviated method used (line 162 – 167), for which, we consider the abbreviated method a strength of the study (line 442 – 448).
Regarding the conclusions, yes, in any case, we are sure of the statements that we have made.
Additionally, we have indicated in the abstract that the associations between variables do not indicate cause and effect (because it is a cross-sectional study). Conversely, we have indicated that the discussion with the literature indicates that VO2max could (of eventually) prevent and mitigate the contagion and effects of COVID-19 (line 54 – 57), which we have substantiated throughout the discussion section.
We have also argued in the discussion section (line 456 – 459) that cause-effect relationships cannot be assumed for the reasons discussed above. We will include this idea again in the conclusion section (line 475 – 476).
Please clarify whether 99.6% of the sample would have suffered a heart attack, before or after the vaccine or having COVID-19, as well as 99.5% of the sample would have suffered a vascular accident. In this case, if they would have suffered such a problem before any of these events, I think it would be important to highlight this as a criterion for the selection of the sample, as we are talking about an almost full percentage.
Dear Reviewer, thank you for your observation. We will correct the absolute and relative frequencies for the mentioned variables, since we write them in reverse order, that is, 99.6% of the sample did NOT suffer a heart attack, and 99.5% did NOT suffer a vascular accident (the same occurred with diabetes) (Table 1).
Reviewer 3 Report
Thank to authors to allow us to obtain information about Chilean, a country not really well documented during pandemic. The investigation is worth of interest while the paper topic remains common, due to the huge amount of peer review articles published on Covid 19 since the last two years.
The use of an approach based on the oxygen consumption is very original and presentation of findings here allows others researchers and specially those from low and middle income countries to replicate this study.
it was a pleasant lecture and is good for publication.
Author Response
Dear Editor, Dr. Federica Valeriani
Special Issue "Sport, Physical Activity and Health at Time of COVID-19"
Response letter to reviewers
In addition to greeting you, we thank the reviewers for the corrections made that have improved the manuscript ijerph-1654180 “Estimated oxygen consumption with abbreviated method and its association with vaccination and PCR tests for COVID-19 from socio-demographic, anthropometric, lifestyle, morbidity outcomes in Chilean adults”. We proceed to respond point by point to said corrections, which are recorded in this response letter and also with track changes in the manuscript.
Cordially
Ivana Leao Ribeiro.
Escuela de Ciencias del Deporte y Actividad Física, Facultad de Salud, Universidad Santo Tomas, Talca PC: 3460000, Chile. ileao@ucm.cl. Phone: +569-712413622.
Marcelo Castillo-Retamal.
Facultad de Ciencias de la Educación, Departamento de Ciencias de la Actividad Física, Universidad Católica del Maule, Talca, PC:3460000, Chile. mcastillo@ucm.cl. Phone: +56 71 298 6414.
Reviewer 3
Thank to authors to allow us to obtain information about Chilean, a country not really well documented during pandemic. The investigation is worth of interest while the paper topic remains common, due to the huge amount of peer review articles published on Covid 19 since the last two years.
The use of an approach based on the oxygen consumption is very original and presentation of findings here allows others researchers and specially those from low and middle income countries to replicate this study.
it was a pleasant lecture and is good for publication.
Dear Reviewer. Thank you for your comments on the information we provide from Chile, on our O2max variable of interest, and for the general assessment of the manuscript. We agree with you that the article will be of interest as long as it contributes something new to the study of COVID-19.
Round 2
Reviewer 2 Report
The authors have made all the corrections requested during the first review process.
This manuscript is a resubmission of an earlier submission. The following is a list of the peer review reports and author responses from that submission.
Round 1
Reviewer 1 Report
The scientific background and the evidences available in literature are not completely reported.
The description of the results of other studies cited in the paper is not fully reported.
The abbreviated method for VO2max estimation is the main weakness of the paper.
The statistical methodology is feasible but the conclusions assumed are excessive in comparison with the parameters included in the analysis.
The limits of the study could be further discussed.
Reviewer 2 Report
The article is well structured and requires a moderate revision of the English language and grammar. It is an interesting paper, clearly written, looking at the association between maximum oxygen consumption (?̇O2max) with vaccination and PCR tests in apparently healthy Chilean adults.
- Abstract: well written
- Introduction: The authors could describe the importance of their research more clearly. LINE 88 is the right COVID-19 and no CO-VID-19
- Materials and Methods: My major concern is with the methods. The enrolment procedure must be better specified, it seems a little confusing who was involved in the survey? How did the authors choose the way used to enroll their sample? How did they avoid the selection bias? What is the reference population? What is the minimum sample considering the reference population and the power of the study? The used questionnaire is taken from a previously published article therefore it is not possible to access information such as the tool validation methodology. What about face validity, reliability, and intelligibility? The authors must report a brief paragraph about this issue and also create a flow diagram of actions.
- Results: I would recommend table 1 placed first and then table 2. The rest is well written.
- Discussion: The authors synthesized the literature and relate the published data to their findings. I suggest emphasizing, a little more, the contribution of the study to the literature. ( https://doi.org/10.22190/FUPES200222042K, https://doi.org/10.2807/1560-7917.ES.2020.25.36.2001542, https://doi.org/10.1101/2021.08.24.21262239) The authors should also write about the Strengths and Limitations of the study.
- Conclusion: 1) M.I. is not explained in the text (Body Mass Index) (DOI 10.26773/jaspe.191007 Evaluation of Municipal Fitness Programs for Women with Low Back Pain) 2) Future scope of this study can be added to stimulate CRF through the practice of physical activity but through MET (Metabolic Equivalent)